# Multimodal Multi-turn Conversation Stance Detection: A Challenge Dataset and Effective Model

Fuqiang Niu
Shenzhen Technology University
Shenzhen, China
nfq729@gmail.com

Zebang Cheng
Shenzhen Technology University
Shenzhen, China
2200411013@stumail.sztu.edu.cn

Xianghua Fu
Shenzhen Technology University
Shenzhen, China
fuxianghua@sztu.edu.cn

Xiaojiang Peng
Shenzhen Technology University
Shenzhen, China
pengxiaojiang@sztu.edu.cn

Genan Dai*
Shenzhen Technology University
Shenzhen, China
daigenan@sztu.edu.cn

Yin Chen
Shenzhen Technology University
Shenzhen, China
chenyin@sztu.edu.cn

Hu Huang*
Peking University
Shenzhen, China
h.huang@pku.edu.cn

Bowen Zhang*
Shenzhen Technology University
Shenzhen, China
zhang_bo_wen@foxmail.com

## Abstract

Stance detection, which aims to identify public opinion towards specific targets using social media data, is an important yet challenging task. With the proliferation of diverse multimodal social media content including text, and images multimodal stance detection (MSD) has become a crucial research area. However, existing MSD studies have focused on modeling stance within individual text-image pairs, overlooking the multi-party conversational contexts that naturally occur on social media. This limitation stems from a lack of datasets that authentically capture such conversational scenarios, hindering progress in conversational MSD. To address this, we introduce a new multimodal multi-turn conversational stance detection dataset (called MmMtCSD). To derive stances from this challenging dataset, we propose a novel multimodal large language model stance detection framework (MLLM-SD), that learns joint stance representations from textual and visual modalities. Experiments on MmMtCSD show state-of-the-art performance of our proposed MLLM-SD approach for multimodal stance detection. We believe that MmMtCSD will contribute to advancing real-world applications of stance detection research.

## CCS Concepts

• **Computing methodologies** → **Language resources**.

*Corresponding authors: Bowen Zhang, Genan Dai and Hu Huang.

## Keywords

Multimodal fusion, Conversational stance detection, Multimodal Large Language Model

**ACM Reference Format:**
Fuqiang Niu, Zebang Cheng, Xianghua Fu, Xiaojiang Peng, Genan Dai, Yin Chen, Hu Huang, and Bowen Zhang. 2024. Multimodal Multi-turn Conversation Stance Detection: A Challenge Dataset and Effective Model. In *Proceedings of the 32nd ACM International Conference on Multimedia (MM '24), October 28–November 1, 2024, Melbourne, VIC, Australia.* ACM, New York, NY, USA, 10 pages. https://doi.org/10.1145/3664647.3681416

## 1 Introduction

Social media enables users to frequently articulate their perspectives on controversial subjects regarding specific entities or topics. Aggregating and analyzing these expressed viewpoints reveals prevailing opinions on divisive topics spanning issues like abortion to epidemic prevention [16]. This wealth of opinionated data holds significant potential for applications in web mining and content analysis. The derived insights can inform various decision-making processes, such as advertising recommendations and presidential elections [26, 34]. Thus, automatically detecting stances in social media has emerged as a key approach to understanding user perspectives on diverse issues [21].

Stance detection aims to determine the expressed opinions or attitudes (favor, against, or none) in content toward specific targets [3, 35]. Conventional machine learning [14, 17] and deep learning [36, 41] approaches have shown significant progress in in processing and analyzing pure text. However, increasingly more social media platforms (e.g. Reddit) enable users to post multimodal content including text (posts and comments), images, etc. This facilitates expressing stances and opinions via multiple modalities. Thus, detecting stances from pure textual content may fail to accurately identify users' real perspectives toward targets. Consequently, multimodal stance detection (MSD) has garnered increased attention in recent research.

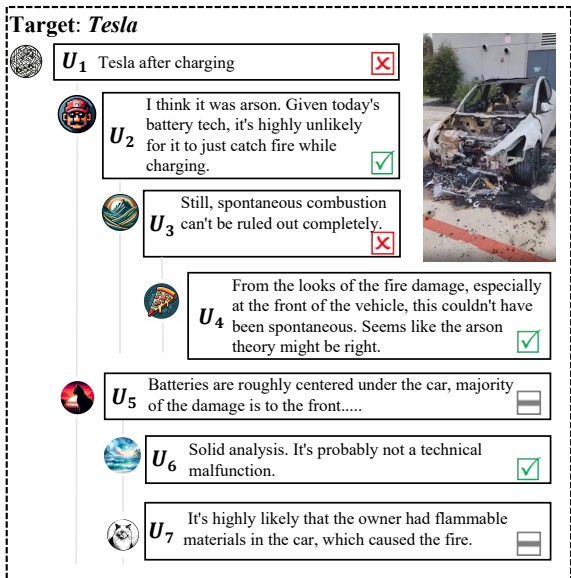

**Figure 1: An example of multimodal multi-turn conversational stance detection, with symbols denoting "favor" (check), "against" (multiplication), and "none" (horizontal line) stances.**

To date, two MSD datasets have been developed and served as benchmarks for MSD tasks, namely MMVax-Stance (MMVax) [40] and MMSD [30]. However, a persistent challenge in real-world social media analysis is that users commonly articulate perspectives through conversational exchanges. Conventional context-free stance detection methods have difficulties accurately predicting stances in such conversational settings. For instance, Figure 1 illustrates a social media discussion. Within this conversational thread, it is difficult to detect the stances of $U4$ towards Tesla without the conversational and image context.

To foster advancements in MSD, we introduce a new multimodal multi-turn conversation stance detection dataset, denoted as **MmMtCSD**[1], in which each example consists of a target, a text, and an image. MmMtCSD contain a total of 21,340 annotated data. Specifically, following [32], we annotated tasks for two targets, "Tesla" and "Bitcoin". Additionally, following [1], we construct tasks where sentences/posts serve as targets, denoted as "Post-T". In the Post-T setting, the targets are diverse, rendering the task more challenging. A salient characteristic of this dataset is statistics annotated by multiple experts showing 66% of conversations are highly related to the image content. This highlights the close interplay between textual and multimedia data. Additionally, the dataset presents two key challenges: first, stance-relevant content mentioned in text is inferable from the multimodal context; second, stance determination heavily relies on contextual cues.

To deal with MmMtCSD, we proposed a novel multimodal large language model stance detection framework (MLLM-SD), which consists of a textual encoder, a visual encoder and a multimodal

fusion module. First, the textual encoder encodes the input conversational history information. Besides, to enhance the association between images and text, we also encode captions of the input images. Second, the visual encoder employs a Vision Transformer (ViT) [15] model to obtain representations of the input images. Third, in the multimodal fusion module, we fine-tuned the LLaMA model using the low-rank adaptation (LoRA) [18] method to integrate information across modalities. The process culminates in matching the output of the large language model (LLM) with the appropriate labels, ensuring a coherent and comprehensive approach to multimodal stance detection.

The main contributions of this paper can be summarized as follows:

- We introduce a challenging MmMtCSD dataset tailored for multimodal stance detection. To the best of our knowledge, this is the first multimodal multi-turn conversational dataset, and the release of MmMtCSD would push forward the research of MSD.
- We propose the MLLM-SD framework, which effectively integrates information across multiple modalities. This framework leverages the comprehension capabilities of LLMs to facilitate a detailed understanding of conversational content coupled with image information.
- We conduct a series of experiments on the MmMtCSD, which substantiated the efficacy of our proposed framework. Additionally, we incorporated various modules into the framework to validate its adaptability.

## 2 Related Work
## 2.1 Stance Detection Datasets

**Sentence-level text-only stance detection datasets.** Over the years, many datasets have been proposed for sentence-level text-only stance detection datasets, serving as benchmarks in this field. The features of these datasets are shown in Table 1. The SemEval-2016 Task 6 (SEM16) dataset [32] pioneered stance detection derived from Twitter and gained widespread adoption. The TSE2020 dataset [20] focused on the 2020 elections, enabling electoral stance analysis. The P-Stance dataset [26] features longer tweets in the political domain, while the expansive WT-WT corpus [9] offers a large labeled resource. Tailored datasets also emerged, including the COVID-19 Stance Detection dataset [16] for pandemic discourse. Most wide-ranging is the VAST dataset [1] for zero-shot stance detection across over a thousand topics. Together, these benchmarks spurred significant research in textual stance detection.

**Conversation-based text-only stance detection datasets.** In real-world scenarios, users typically express perspectives in a conversational manner. Consequently, conversational stance detection, which identifies stances within conversation threads, has garnered increasing research attention recently. The SRQ dataset [38] pioneered stance detection for comment data, but was constrained to single-turn replies and shallow conversation depth, rendering it inapplicable for real commenting scenarios. The CANT-CSD dataset [25] aimed to address stance detection in multi-turn dialogues, providing a deeper commenting corpus. However, CANT-CSD is in Cantonese, with still limited conversation depth. The CTSDT [27] dataset relies mainly on automated annotations rather

**Table 1: Comparison of different stance detection datasets. Here, *n* represents the number of posts acting as targets, and therefore, there is no fixed quantity.**

| Type | Textual | | | | | | Multimodal | | |
|---|---|---|---|---|---|---|---|---|---|
| Classif. Task | Textual classification | | | | | | Multimodal classification | | |
| Work | SEM16, P-stance COVID-19-Stance WT-WT, TSE2020 | VAST | SRQ | CANT-CSD | CTSDT | MT-CSD | MMVax | MMSD | Our work |
| Target Number | < 6 | $n$ | 4 | 1 | 1 | 5 | 1 | 5 | $2 + n$ |
| English | ✔ | ✔ | ✔ | ✘ | ✔ | ✔ | ✔ | ✔ | ✔ |
| Conversation | ✘ | ✘ | ✔ | ✔ | ✔ | ✔ | ✘ | ✘ | ✔ |
| Multi-turn | ✘ | ✘ | ✘ | ✔ | ✔ | ✔ | ✘ | ✘ | ✔ |

than manual labeling. The MT-CSD dataset [34] expanded conversation turns more extensively for CSD, employing English to enable broader applicability.

**Multimodal stance detection dataset.** Weinzierl and Harabagiu [40] pioneered the creation of the first multimodal stance detection dataset MMVax specifically for COVID-19, comprising 11,300 instances. Subsequently, Liang et al. [30] expanded existing text-based stance detection datasets (e.g. TSE2020, WT-WT) by incorporating image content and re-annotation to construct the larger MMSD multimodal stance detection dataset, totaling 17,544 annotated instances. However, to the best of our knowledge, prior work has not explored multimodal stance detection in conversational threads, motivating our current research.

## 2.2 Stance Detection Approaches

In recent years, various approaches based on traditional machine learning and deep learning have been proposed to address stance detection for specific targets [2]. The task settings can generally be categorized into in-target[24], cross-target [11, 29, 39], and zero-shot settings [1, 28, 42]. Current methods can typically be classified into fine-tuning-based approaches and LLM-based approaches. Fine-tuning-based methods involve adding a fully connected layer to the [CLS] token of a pre-trained model (such as BERT) and fine-tuning the model for the stance detection task [5]. Recently, LLMs have demonstrated remarkable capabilities across diverse applications, owing to their inherent semantic capabilities [6, 7]. The semantic understanding of LLMs presents exciting opportunities for stance detection [23]. Most LLMs can easily perform stance prediction via zero-shot prompting by users, significantly enhancing usability [12].

## 3 Dataset Construction

In this section, we provide a detailed overview of the creation process and unique attributes of our MmMtCSD dataset comprising 21,340 texts and images.

**Table 2: The number of data items for each target.**

| Target | Tesla | Bitcoin | Post-T | Total |
|---|---|---|---|---|
| Post | 774 | 637 | 463 | 1,874 |
| Comment | 17,936 | 18,038 | 20,753 | 56,727 |

### 3.1 Data Collection

To acquire authentic conversational data from social media, we leveraged Reddit[2], one of the largest and most extensive online forums, ensuring the richness and authenticity of the collected MmMtCSD data. We accessed the data through Reddit's official API[3]. To obtain topics with sufficient discussion and high relevance, during the data collection process, we gathered Reddit posts and associated popularity metrics, such as upvotes and comment counts. A manual review was conducted to assess the relevance of the posts (post texts and images) to the given targets, ensuring that the collected posts were highly pertinent and featured sufficiently in-depth comments to facilitate dataset annotation. Subsequently, we collected comments for each selected post. The resulting dataset encompassed relevant posts, associated discussions, and comments, providing a comprehensive overview of conversations centered around the specified targets. The selected targets for this dataset included "Tesla" and "Bitcoin", and we additionally constructed posts as targets.

### 3.2 Data Preprocessing

To ensure the high quality of this MmMtCSD dataset, we implemented several rigorous preprocessing steps:

- High Relevance to Target: To ensure strong correlation between post content and the specified target, a rigorous evaluation process involving two reviewers was implemented. Only posts deemed highly relevant by both reviewers were included in the dataset.
- Minimum 100 Comments per Post: A minimum threshold of 100 comments per post was established to ensure substantial attention and discourse. Posts with fewer comments were excluded to maintain conversational depth and complexity, essential for capturing nuanced stances in multi-turn exchanges.
- Appropriate Text Length: Constraints were imposed on post length to maintain data quality. Posts were required to be between 15 and 150 words. Posts shorter than 15 words were considered too simplistic, while those longer than 150 words often contained redundant expressions.
- Excluding Non-English Posts: To construct an all-English dataset, non-English posts were systematically removed to maintain language consistency. Multilingual stance detection remains a potential area for future exploration.

---

[2]https://www.reddit.com
[3]https://www.reddit.com/dev/api

**Table 3: Label distribution of the MmMtCSD dataset, with "Vision-related" indicating the number of data entries related to images.**

| Target | Samples and Proportion of Labels | | | | | | | Vision-Related | |
|---|---|---|---|---|---|---|---|---|---|
| | Against | % | Favor | % | None | % | Total | Count | % |
| **Tesla** | 2,211 | 35.10 | 2,531 | 40.17 | 1,558 | 24.73 | 6,300 | 3,308 | 40.56 |
| **Bitcoin** | 1,284 | 15.76 | 4,550 | 55.84 | 2,314 | 28.40 | 8,148 | 4,529 | 71.89 |
| **Post-T** | 2,008 | 29.14 | 3,255 | 47.23 | 1,629 | 23.64 | 6,892 | 6,246 | 90.63 |
| **Total** | 5,503 | 25.79 | 10,336 | 48.43 | 5,501 | 25.78 | 21,340 | 14,083 | 65.99 |

**Table 4: Statistics of the MmMtCSD dataset. Here, WC is short for word count.**

| Instance | Avg. WC | Depth | Number |
|---|---|---|---|
| **Post** | 39.81 | 1 | 955 (4.5%) |
| **Comment** | 27.18 | 2 | 4,605 (21.58%) |
| | 29.96 | 3 | 6,076 (28.47 %) |
| | 33.13 | 4 | 4,733 (22.18 %) |
| | 39.23 | 5 | 3,230 (15.14%) |
| | 47.20 | 6 | 1,741 (8.16%) |

Furthermore, to ensure the inclusion of multimodal content, we filtered the posts to remove those lacking multimodal elements. After completing these preprocessing steps, we obtained 107,249, 112,081, and 140,129 data instances for the Bitcoin, Tesla, and Post targets, respectively. The resulting data distribution is summarized in Table 2.

## 3.3 Data Annotation and Quality Assurance

To enable meticulous annotation accounting for conversational context, we developed a custom system requiring reviewers to thoroughly examine all preceding multimodal content (text and images) prior to assigning stance labels. This framework was tailored to streamline high-quality annotation of multimodal conversation datasets. During the annotation phase, clear guidelines were provided to annotators, instructing them to label each comment as "*against*", "*favor*", or "*none*" to reflect their assessed stance. In addition to these stance labels, annotators were also tasked with assessing and marking the relevance of the data, determining whether it was related to the images in the post. This comprehensive annotation process ensured a thorough and contextually informed labeling of the dataset.

We recruited seven researchers with expertise in natural language processing (NLP) to undertake the data annotation task. To validate the consistency and reliability of the annotations, two preliminary rounds of pilot annotation were conducted, followed by a review from three additional expert annotators to confirm the ability of each annotator to accurately execute the task. In the main annotation phase, we ensured that at least two annotators independently reviewed each data instance. Discrepancies between the initial annotators were resolved by involving a third annotator to evaluate the disputed instances, with the final stance determination achieved through a majority vote. This rigorous process not only

secured the reliability of the dataset but also leveraged collective expertise to enhance the precision of the stance categorization.

After completing the annotation process, we calculated the *kappa* statistic [8] to assess the consistency among annotators. Following Li et al. [26], we specifically focused on the "Favor" and "Against" categories to determine the kappa statistic values. The *kappa* scores for Bitcoin, Tesla, and Post-T targets were found to be 0.72, 0.81, and 0.68, respectively, indicating a substantial level of agreement among the annotators and attesting to the reliability of the annotation process for our dataset.

## 3.4 Data Analysis

Table 3 presents the statistical data for our MmMtCSD dataset. The final annotated dataset encompasses 21,340 instances, with 14,083 of these instances, or 66%, being related to image content, underscoring the significance of multimodal data inclusion. Table 4 illustrates the distribution of instances across different conversational depths, including the number of words at each depth. Subsequently, we partitioned the dataset into training, validation, and test sets for all targets in a 70/15/15 ratio, ensuring a balanced representation for comprehensive evaluation and analysis.

## 4 Methodology

This section provides a detailed description of our proposed MLLM-SD framework. Given a post and its accompanying comment text $U$, along with images $V$ included in the post, the aim of MLLM-SD is to identify the stance label $y$ towards a specific target $t$. To effectively leverage both textual and visual information, we devised a targeted multimodal prompting approach, which is utilized to prompt the LLM for learning multimodal stance features.

The architecture of our MLLM-SD is depicted in Figure 2, comprises three primary components: (1) A textual encoder encoding textual prompts and content; (2) A visual encoder processing and encoding images with visual prompts; and (3) A multimodal fusion module where the LLM integrates information across modalities. The process culminates in matching the LLM's output with the appropriate labels, ensuring a cohesive and comprehensive approach to multimodal stance detection.

## 4.1 Textual Encoder

The textual encoder aims to construct prompt templates for different types of text inputs, serving as the input for the subsequent LLM in the multimodal fusion stage. Specifically, within the textual encoder, in addition to the conversation history text data, we incorporate image caption designed to enhance the association between visual

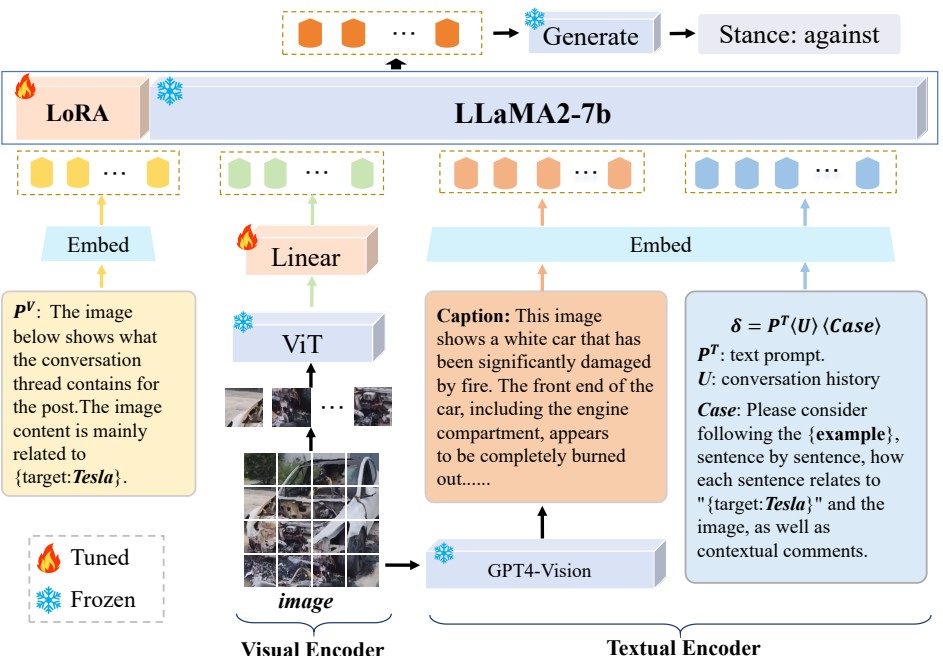

Figure 2: The architecture of our MLLM-SD framework.

and textual information. Notably, we generate prompt templates for the conversational data, while for the image, we employ GPT4-Vision[4] to generate textual summaries.

Specifically, to facilitate LLM understanding of multi-turn conversation stance detection, we first followed the LLaMA-2 conversation template design, specifying [*stance detection*] as the task definition within the template. We then incorporated the conversation text $U = \langle u_1, u_2, \ldots, u_n \rangle$, comprising a sequence of $n$ utterances $u_i = \langle w_{i,1}, w_{i,2} \ldots, w_{i,j} \rangle$  $(\forall j = 1, \ldots, l_i)$ representing posts or comments as the $U$.

Furthermore, inspired by the chain-of-thought (CoT) method [4], to enhance the model's task performance, we define a one-shot prompt template $\delta$, which packages the $P^T$, $U$, and provides an example, denoted as *Case*, as a one-shot prompt, constructing a new input format.

---

$P^T$: The following is a conversation on social media based on a post. All comments are responses to the content of the post, and each comment replies to the previous one. There are three stances [favor, against, none]. Choose one of the three stances to express {**name**: $u_i$}'s stance towards "{**target**}".

---

The $P^T$, conversation history $U$, and one-shot CoT structure together form the one-shot prompt $\delta$:

$$\delta = \langle P^T \rangle \langle U \rangle \langle Case \rangle \tag{1}$$

Second, to allow better modeling of relationships between visual and textual information, we generated image descriptions using GPT4-Vision as *Caption*. The caption and one-shot prompt together

---

[4]https://openai.com/research/gpt-4v-system-card

constitute the textual modality input, denoted as

$$\gamma^T = [stance\ detection] \langle Caption \rangle \langle \delta \rangle. \tag{2}$$

Finally, we utilize a tokenizer from LLaMA to encode $\gamma^T$ into $\Gamma^T$.

## 4.2 Visual Encoder

Following [13], we first divide the image $V$ into a sequence of flattened 2D patches. These patches are then encoded into a linear sequence of embeddings which serve as input to the ViT model. We embed each patch into a embedding $x_p^i$. Next, we embed each patch into a vector with positional encoding $\mathbf{E}_{pos}$ through an embedding projection $\mathbf{E}$:

$$v_0 = [x_{class}; x_p^1\mathbf{E}; x_p^2\mathbf{E}; \ldots; x_p^N\mathbf{E}] + \mathbf{E}_{pos} \tag{3}$$

where $v_0$ is the embedding for the input image. Subsequently, we use the pre-trained Vision Transformer model ViT [15] to encode the input image for learning visual stance features. The resulting feature vector $y$ is processed by a linear transformation layer to adjust its feature dimensions to $d_v = 4096$, resulting in the image features $\Gamma^V$ This transformation is critical for aligning the visual feature dimensions with the requirements of our model's architecture.

## 4.3 Multimodal Fusion

In the multimodal fusion stage, we employed LoRA method to fine-tune the LLaMA2-chat (7B) [37] model, enabling it to effectively process text and images, manage multimodal inputs, and generate natural language responses to the proposed queries. To enhance the model's understanding of the input content, we utilized $P^V$ (Figure 2 $P^V$) as a prefix for the input image, representing the structural form of the data. This prefix helps define the nature of the input and

assists the model in grasping the structure of the multimodal data. The input architecture for the model is as follows:

$$[INST]P^V \langle Img \rangle \ \Gamma^V \langle /Img \rangle \ \Gamma^T [/INST] \quad (4)$$

where $[INST]$ denotes the user role, and $[/INST]$ signifies the assistant role. The user input is organized into two segments: the first part comprises the image features ($\Gamma^V$), and the second part consists of the textual instruction input ($\Gamma^T$). Following the model's output generation, we engage in similarity matching to ascertain the stance conveyed by the output content.

## 5 Experimental Setup

In this section, we detail the baseline models utilized in our experiments, encompassing the experimental setup, which includes evaluation metrics, implementation details, and tests of the applicability of the MLLM-SD framework.

### 5.1 Baseline Methods

We conduct extensive experiments utilizing state-of-the-art stance detection methods, broadly categorized into text-only and multimodal approaches. These experiments were designed to evaluate the performance of these methods in various stance detection scenarios, allowing for a comprehensive analysis of their effectiveness and the additional value provided by integrating multimodal data in the stance detection process.

**Text-only baselines.** *Fine-tuning based methods*: (1) the pre-trained **BERT** [10] is fine-tuned on the training data; (2) the pre-trained **RoBERTa** [31] represents an enhancement over BERT, utilizing larger batch sizes and more data for training; (3) **KEPrompt** [19] uses an automatic verbalizer to automatically define the label words; (4) **Branch-BERT** [25] utilizes a CNN to extract important n-grams features incorporating contextual information in conversation threads. (5) **GLAN** [34] architecture adopts a three-branch structure to address the intricacies of conversational dynamics comprehensively. *LLM-based methods*: For the text stance detection task, we followed the prompting method proposed by Lei et al. [22] and utilized **LLaMA-70b**[5], **Claude-3**[6], **ChatGPT** (gpt-3.5[7] and gpt-4[8]) as the comparison method.

**Multimodal baselines.** Three methods for multimodal modeling are employed as baselines: (1) **BERT+ViT** [30] utilizes BERT for textual encoding and ViT [13] for visual encoding. (2) **TMPT** [30] employs targeted prompts supplied to both the pre-trained language model and the pre-trained visual model. (3) **GPT4-Vision**[4] represents a vision-enhanced large model approach, integrating the capabilities of GPT-4 with visual processing to understand and analyze multimodal data for stance detection.

### 5.2 Experimental Settings

**Evaluation metrics**. We adopt F1-avg as the evaluation metric to evaluate the performance of stance detection methods, consistent with the approaches in [26] and [33]. F1-avg represents the average F1 score computed for the "against" and "favor" stances, denoted as F1-against and F1-favor, respectively.

---

[5] https://huggingface.co/meta-llama/Llama-2-70b-chat-hf
[6] https://www.anthropic.com/api
[7] https://platform.openai.com/docs/models/gpt-3-5
[8] https://platform.openai.com/docs/models/gpt-4-and-gpt-4-turbo

**Implementation details**. During the training regimen of the MLLM-SD framework, the visual backbone is maintained in a static (frozen) state to preserve pre-trained visual features. The emphasis is placed on training the linear projection layer and fine-tuning the language model efficiently using LoRA. Through LoRA, the query and value weight matrices ($W_q$ and $W_v$, respectively) are fine-tuned, enabling model adaptation with minimal parameter updates. In our implementation, we specified the rank for LoRA as 64, striking a balance between adaptability and computational efficiency. Additionally, the model was consistently trained using an image resolution of 448x448 across all training phases, ensuring uniformity in visual data processing.

## 6 Experimental Results

In this section, we perform comprehensive experiments on our MmMtCSD dataset. Concretely, we present model comparisons in both in-target and cross-target setups. Notably, the reported results are averages obtained from three distinct initial runs.

### 6.1 In-Target Stance Detection

We first report the experimental results on the MmMtCSD dataset in the in-target setup, as shown in Table 5, where the training and testing sets share identical targets. From the results, we make the following observations: First, the models built on multimodal inputs consistently outperform text-only models, highlighting advantages and necessity of multimodal inputs. Secondly, the simple concatenation of different modal LLMs does not lead to satisfactory performance on MMSD, with BERT+ViT achieving only 61.15%, while TMPT scored 64.47%, in evaluations across *Tesla* and *Bitcoin* targets. This phenomenon could be attributed to the difficulty in capturing high-level semantics within and across modalities through simple concatenation. Third, our MLLM-SD outperforms all baseline models on the MmMtCSD dataset. The significance tests comparing MLLM-SD to BERT+ViT and TMPT reveal that MLLM-SD exhibits a statistically significant improvement across most evaluation metrics (with a $p$-value of $< 0.05$). Fourth, even state-of-the-art stance detection methods, exemplified by MLLM-SD, exhibit an F1 score of only 71.85%, highlighting the persistent challenges in conversational stance detection.

### 6.2 Cross-Target Stance Detection

We undertook a series of cross-target experiments on the MmMtCSD dataset. The stance detection models are initially trained and validated on a source target and subsequently tested on a destination target. Our experimental design encompasses all available targets. The results of cross-target multimodal stance detection are reported in Table 6. It can be seen that the LLMs achieve superior performance due to the need for detecting stances on unseen targets. This may be attributed to the powerful cross-target learning capability of LLMs. For our proposed MLLM-SD method, it achieves optimal results compared to large model-based methods on the Tesla and Bitcoin targets, while also outperforming all non-LLM baselines. This demonstrates the potential of our method in multimodal stance detection. When the stance targets become more diverse, MLLM-SD achieves sub-optimal results on the cross-target task (Post-T). This may be attributed to the reason that the training data in this setting

**Table 5: In-target experimental results (%) on the MmMtCSD dataset, with the best performance in each group highlighted in bold and the second best underlined.**

| MODALITY | METHOD | Tesla | | | Bitcoin | | | Post-T | | |
|---|---|---|---|---|---|---|---|---|---|---|
| | | F1-against | F1-favor | F1-avg | F1-against | F1-favor | F1-avg | F1-against | F1-favor | F1-avg |
| Text-only | BERT | 51.62 | 60.50 | 56.06 | 38.78 | 71.69 | 55.24 | 60.51 | 73.23 | 66.87 |
| | RoBERTa | 54.24 | 63.96 | 59.10 | 41.06 | 69.98 | 55.52 | 60.94 | 72.76 | 66.85 |
| | KEPrompt | 50.97 | 59.72 | 55.35 | 33.58 | 70.06 | 51.82 | 55.34 | 71.82 | 63.58 |
| | Branch-BERT | 51.92 | 60.82 | 56.37 | 39.61 | 70.66 | 55.14 | 55.91 | 69.70 | 62.81 |
| | GLAN | 53.06 | 61.40 | 57.23 | 42.65 | 70.32 | 56.49 | 59.71 | 76.13 | 67.92 |
| | LLaMA 2-70b | 55.03 | 65.67 | 60.35 | 48.49 | 75.00 | 61.75 | 58.20 | 71.77 | 64.99 |
| | Claude-3 | 49.09 | 64.13 | 56.61 | 40.63 | 71.35 | 55.99 | 42.15 | 50.51 | 46.33 |
| | ChatGPT(gpt-3.5) | 56.32 | 65.35 | 60.84 | 42.55 | 67.00 | 54.78 | 61.67 | 70.95 | 66.31 |
| | ChatGPT(gpt-4) | 54.12 | 59.70 | 56.91 | 57.26 | 75.98 | 66.62 | 61.15 | 61.93 | 61.54 |
| Multi-modal | BERT+ViT | 54.25 | 62.46 | 58.36 | 40.78 | 72.74 | 56.76 | 74.50 | 62.18 | 68.34 |
| | TMPT | 56.86 | 62.57 | 59.72 | 52.57 | 73.23 | 62.90 | 72.46 | 69.12 | 70.79 |
| | GPT4-Vision | 56.65 | 65.96 | 61.31 | 49.45 | 73.11 | 61.28 | 69.24 | 70.83 | 70.04 |
| | **MLLM-SD** | **62.64** | 67.13 | **64.89** | **65.21** | **77.35** | **71.28** | **79.56** | **79.23** | **79.40** |
| | *w/o Caption* | 57.53 | **68.41** | 62.97 | 63.26 | 75.43 | 69.35 | 72.74 | 75.13 | 73.94 |
| | *w/o Cot* | 60.13 | 66.63 | 63.38 | 62.63 | 76.82 | 69.73 | 74.99 | 76.34 | 75.67 |
| | *w/o Cot & Caption* | 58.35 | 67.35 | 62.85 | 60.84 | 76.13 | 68.49 | 70.75 | 72.40 | 71.58 |

**Table 6: Comparison of different models for cross-target stance detection, where the methods based on LLMs involves testing through direct questioning.**

| MODALITY | METHOD | Tesla | | | Bitcoin | | | Post-T | | |
|---|---|---|---|---|---|---|---|---|---|---|
| | | F1-against | F1-favor | F1-avg | F1-against | F1-favor | F1-avg | F1-against | F1-favor | F1-avg |
| Text-only | BERT | 40.23 | 37.76 | 39.00 | 36.43 | 46.37 | 41.40 | 30.97 | 41.56 | 36.27 |
| | RoBERTa | 42.17 | 38.58 | 40.38 | 39.84 | 48.29 | 44.07 | 35.87 | 37.12 | 36.50 |
| | Branch-BERT | 42.24 | 51.96 | 47.10 | 31.41 | 60.92 | 46.17 | 39.51 | 43.28 | 41.39 |
| | GLAN | 43.97 | 50.75 | 47.36 | 48.97 | 50.45 | 49.71 | 39.39 | 45.85 | 42.62 |
| | LLaMA-70b | 49.26 | 55.86 | 52.56 | 41.08 | 68.44 | 54.76 | 55.69 | 61.19 | 58.44 |
| | Claude-3 | 52.33 | 54.23 | 53.28 | 36.00 | 59.45 | 47.73 | 44.61 | 41.07 | 42.84 |
| | ChatGPT(gpt-3.5) | 53.10 | 50.01 | 51.56 | 37.03 | 54.72 | 45.88 | 51.36 | 32.46 | 41.91 |
| | ChatGPT(gpt-4) | 52.12 | 51.13 | 51.63 | 45.37 | 62.12 | 53.75 | 46.43 | 58.42 | 52.43 |
| Multi-modal | BERT+ViT | 33.43 | 35.05 | 34.24 | 32.73 | 38.63 | 35.68 | 36.92 | 33.47 | 35.20 |
| | TMPT | 37.65 | 41.75 | 39.70 | 34.12 | 35.23 | 34.68 | 31.97 | 35.02 | 33.50 |
| | GPT4-Vision | 55.23 | 57.32 | 56.28 | 47.75 | **70.23** | 58.99 | **59.64** | **66.42** | **63.03** |
| | **MLLM-SD** | **56.24** | 58.21 | **57.23** | **56.57** | 62.29 | **59.43** | 53.04 | 64.34 | 58.69 |
| | *w/o Caption* | 54.88 | 54.77 | 54.83 | 48.74 | 54.07 | 51.41 | 53.90 | 57.22 | 55.56 |
| | *w/o Cot* | 55.14 | **58.38** | 56.76 | 51.39 | 60.69 | 56.04 | 55.71 | 58.22 | 56.97 |
| | *w/o Cot & Caption* | 54.60 | 54.76 | 54.68 | 46.71 | 54.18 | 50.45 | 50.92 | 62.12 | 56.52 |

originates from only two specific domains, Tesla and Bitcoin, which could negatively impact the model's prediction capability for such a broad range of stance targets. In future research, exploring techniques such as data augmentation could be a promising direction to enhance the model's performance on cross-target multimodal stance detection.

## 6.3 Impact of Conversation Information

To validate the impact of conversational context on stance detection, we constructed a comparison between single sentence and conversation history based models using LLaMA2-70b for both text-only and multimodal tasks. The results are illustrated in Figure 3.

The findings demonstrate that leveraging conversation history can significantly improve the accuracy of stance detection compared to relying solely on single-sentence inputs. Notably, in the multimodal stance detection scenario, the influence of contextual information on the results is more pronounced than in the text-only setting. This further accentuates the pivotal importance of incorporating comment conversation history in multimodal stance analysis.

## 6.4 Impact of Conversation Depth

This analysis aims to examine the performance of different stance detection models across varying conversation depths. The results

**Table 7: Performance evaluation showing F1-avg comparison of different models across instances with depths 1, 2-4, and 5-6, considering conversation history. For "Post-T", the post (depth 1) is considered the target, and thus the depth starts from 2.**

| Target depth | Tesla | | | Bitcoin | | | Post-T | | |
|---|---|---|---|---|---|---|---|---|---|
| | 1 | 2-4 | 5-6 | 1 | 2-4 | 5-6 | 2 | 3-4 | 5-6 |
| BERT | 45.81 | 63.46 | 52.30 | 64.29 | 53.47 | 53.35 | 61.24 | 66.52 | 69.49 |
| RoBERTa | 61.82 | 59.24 | 51.25 | 55.29 | 57.69 | 52.24 | 61.06 | 68.83 | 67.78 |
| KEPrompt | 53.24 | 53.17 | 63.76 | 49.58 | 52.57 | 53.12 | 58.13 | 63.11 | 69.54 |
| Branch-BERT | 55.38 | 61.96 | 56.13 | 56.00 | 56.36 | 49.68 | 63.69 | 58.40 | 67.90 |
| GLAN | 44.04 | 57.96 | 59.12 | 50.56 | 55.96 | 58.50 | 65.14 | 70.10 | 73.93 |
| LLaMA-70b | 72.53 | 62.94 | 69.26 | 70.14 | 62.54 | 62.08 | 64.46 | 59.68 | 67.27 |
| Claude-3 | 60.92 | 55.93 | 67.70 | 56.25 | 54.30 | 59.42 | 42.26 | 44.38 | 50.98 |
| ChatGPT(gpt-3.5) | 62.28 | 58.52 | 68.16 | 60.51 | 53.46 | 53.38 | 62.37 | 65.16 | 64.71 |
| ChatGPT(gpt-4) | 63.40 | 58.39 | 63.12 | 73.65 | 64.62 | 73.21 | 41.06 | 69.44 | 65.94 |
| BERT+ViT | 69.67 | 59.25 | 51.65 | 47.05 | 51.19 | 56.48 | 65.34 | 59.75 | 73.70 |
| TMPT | 53.12 | 59.96 | 60.34 | 52.31 | 61.47 | 59.89 | 66.64 | 61.25 | 71.25 |
| GPT4-Vision | 58.34 | 61.04 | 63.75 | 51.76 | 63.45 | 62.35 | 63.72 | 62.28 | 70.72 |
| **MLLM-SD** | **64.68** | 63.11 | **65.12** | 70.10 | 68.21 | **74.89** | **75.85** | **77.42** | **84.85** |
| *w/o Caption* | 60.70 | **64.16** | 64.50 | 69.72 | 66.44 | 72.50 | 74.57 | 69.51 | 78.29 |
| *w/o Cot* | 61.97 | 60.18 | 63.68 | **71.73** | **68.99** | 69.21 | 74.38 | 72.50 | 75.48 |
| *w/o Cot & Caption* | 59.79 | 63.21 | 63.62 | 68.98 | 65.51 | 68.49 | 72.12 | 68.57 | 71.51 |

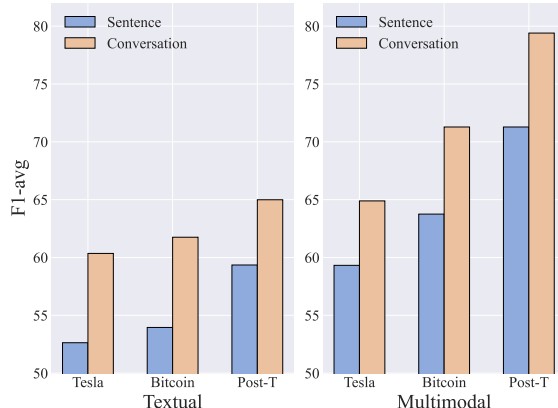

**Figure 3: Comparison of single-sentence and conversation results: Textual display represents the experimental outcomes for LLaMA2-70b, while multimodal display showcases the results for MLLM-SD.**

for different conversation depths are reported in Table 7. The findings indicate that our model achieves enhanced performance across all measured depths. Notably, with the depth 5-6, our method demonstrates a more significant improvement in overall stance detection performance. This could be attributed to our innovative introduction of image captioning, which aids in capturing the contextual information between the text and images.

## 6.5 Ablation Study

To analyze the impact of different modules in our proposed MLLM-SD framework, we conduct an ablation study and report the results

in Tables 5-7. Note that the removal of caption (w/o *Caption*) sharply degrades the performance, which verifies the significance and efficacy of leveraging LLMs to summarize image content. This approach facilitates the model's comprehension of visual information and its association with the conversational text. In addition, the removal of the CoT process (w/o *CoT*) results in a substantial performance decline, suggesting that the CoT mechanism enhances the model's capability to interpret and analyze intricate instances. Furthermore, the concurrent removal of both CoT and Caption (w/o *CoT & Caption*) components results in substantial degradation in performance, underscoring the critical role these elements play in enhancing the overall effectiveness of the framework.

## 7 Conclusion

This paper introduces MmMtCSD, an extensive English multimodal conversational stance detection benchmark, specifically designed to emphasize multi-turn conversations. MmMtCSD addresses critical challenges in the multimodal stance detection task, aiming to bridge the gap between research and real-world applications. We propose a novel MLLM-SD framework that learns joint stance representations from textual and visual modalities. We conduct comprehensive experiments on our MmMtCSD dataset, and the experimental results demonstrate that MLLM-SD achieves superior performance on the MmMtCSD benchmark. Furthermore, extensive experimental findings underscore that MmMtCSD poses a more formidable challenge compared to existing benchmarks. This highlights substantial opportunities for advancements and innovations in stance detection. In future work, we plan to integrate linguistic knowledge and user information to further enhance the performance of multimodal conversational stance detection.

## 8 Acknowledgements

This research is supported by National Nature science Foundation of china (No.62306184, 62176165), Natural Science Foundation of Top Talent of SZTU (grant no. GDRC202320, GDRC202131, GDRC202133), Shenzhen Science and Technology Program (Grant No.RCBS20231211090548077) and the Research Promotion Project of Key Construction Discipline in Guangdong Province (2022ZDJS112).

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
