# OpenReview forum: "Multimodal Multi-turn Conversation Stance Detection: A Challenge Dataset and Effective Model"
_acmmm.org/ACMMM/2024/Conference — MM2024 Oral_

### Official Review · Reviewer_Hd8L · 2024-04-27
**Good paper, but has important weakness to address**

**Rating:** 5
**Confidence:** 4

**Review:**

Advantages:

- The authors introduce an intriguing Multimodal Multi-turn Conversation Stance Detection dataset and task, which closely resembles real-world scenarios, setting it apart from previous stance detection research.
- Their novel approach utilizing multi-modal LLM for stance detection demonstrates effectiveness.
- The paper exhibits strong writing quality and clarity.

Despite these strengths, I didn't assign a higher rating due to the following weaknesses:

- Primary Weakness: The absence of comparison with the significant Multi-turn Conversation Stance Detection paper from ICDM 2023, "Contextual Target-Specific Stance Detection on Twitter: New Dataset and Method," in terms of both dataset creation and method effectiveness. The paper proposed one of the latest Multi-turn Conversation Stance Detection dataset, and also the SOTA method that should be considered a strong baseline. https://ieeexplore.ieee.org/abstract/document/10415673
- The authors need to emphasize the importance of "Multimodal Multi-turn Conversational Stance Detection." Specifically, they should clarify the necessity of both conversational context and visual information simultaneously. The current evidence, such as the relevance of images in 66% of conversations and the existence of cases requiring both types of information concurrently, appears insufficient.
- Unanticipated experimental results: Tables 6 & 7 show GPT-3.5 outperforming GPT-4 when targeting "Post-T," which seems unusual.
The authors should assess the robustness of their methods to different pre-given prompts, considering the significant impact prompts can have on model performance.

Additional Comments:

- The authors might consider extending their research to encompass multi-language support.

**Summary:**

The authors propose a new task and a new dataset named Multimodal Multi-turn Conversation Stance Detection. They also proposed an MLLM based method to address this task.

**Strengths:**

I think this is an interesting work of good quality. The strengths are listed as follows:.
- The authors introduce an intriguing Multimodal Multi-turn Conversation Stance Detection dataset and task, which closely resembles real-world scenarios, setting it apart from previous stance detection research.
- Their proposed approach, utilizing multi-modal LLM for stance detection, looks relatively novel and demonstrates effectiveness.
- Most of the evaluations are solid and comprehensive.
- The paper exhibits strong writing quality and clarity.

**Limitations:**

Despite these strengths, I didn't assign a higher rating due to the following weaknesses:

- Primary Weakness: The absence of comparison with the significant Multi-turn Conversation Stance Detection paper from ICDM 2023, "Contextual Target-Specific Stance Detection on Twitter: New Dataset and Method," in terms of both dataset creation and method effectiveness. The paper proposed one of the latest Multi-turn Conversation Stance Detection dataset, and also the SOTA method that should be considered a strong baseline. https://ieeexplore.ieee.org/abstract/document/10415673
- The authors need to emphasize the importance of "Multimodal Multi-turn Conversational Stance Detection." Specifically, they should clarify the necessity of both conversational context and visual information simultaneously. The current evidence, such as the relevance of images in 66% of conversations and the existence of cases requiring both types of information concurrently, appears insufficient.
- Unanticipated experimental results: Tables 6 & 7 show GPT-3.5 outperforming GPT-4 when targeting "Post-T," which seems unusual.
The authors should assess the robustness of their methods to different pre-given prompts, considering the significant impact prompts can have on model performance.

Additional Comments:

- The authors might consider extending their research to encompass multi-language support.

**Suitability:**

2

---

### Official Review · Reviewer_QNXi · 2024-05-19

**Rating:** 5
**Confidence:** 3

**Summary:**

I would like to give a **weak accept** to this paper under review. The rest of my comments are organized into strengths, weaknesses, and suggestions for improvement.

**Strengths:**

- The authors successfully **identify a research gap** due to the absence of datasets that capture multi-party conversational contexts in multimodal stance detection. To address this, they introduce a new stance detection dataset, MmMtCSD.
- The paper proposes **a novel framework**, MLLM-SD, which integrates large language models with visual and textual data to effectively determine stances in social media conversational threads.

**Limitations:**

- **Missing Model Training Details**: The paper lacks a detailed section on model training and parameter settings, mentioning only the use of LoRA to fine-tune LLaMA at line 153 without further details.
- **Limited Cross-Domain Applicability**: The dataset focuses primarily on two specific domains (Tesla and Bitcoin), potentially limiting the generalizability of the model to other contexts or topics.
- **Incomplete Model Comparisons**: The comparisons are limited and may be biased; for instance, the paper lacks comparative analysis with other well-known large language models—such as Flan-T5, Vicuna, and others—after fine-tuning them on the same dataset, and disparities in model sizes could also lead to unfair comparisons.

**Suitability:**

3

---

### Official Review · Reviewer_A95K · 2024-05-20

**Rating:** 4
**Confidence:** 2

**Summary:**

This paper introduces a new multimodal multi-turn conversational stance detection dataset to address the limitation of existing MSD studies ignoring the multi-party conversational contexts that naturally occur on social media. Besides, this paper proposes a novel multimodal large language model stance detection framework to derive stances from this new dataset.

**Strengths:**

1 The overall writing of this paper is clear, the method level is clear and easy to understand.

2 This paper proposes a new multimodal stance detection dataset, which considers the multi-party conversational contexts in social media.

3 This paper leverages the comprehension capabilities of LLMs to detect stance across multiple modalities.

4 The experiment is sufficient and the related analysis is comprehensive.

**Limitations:**

1 For dataset:

(1) Why Tesla and Bitcoin as targets? In addition, Post-T as a target is not very clear.

(2) Vision-Related in Table 3 reaches 65.99%, then how to deal with irrelevant images?

(3) Does each target have just one long conversation? Or a combination of multiple conversations? Is tagging an utterance for each utterance in a conversation?

(4) Different targets were evaluated in the experiment, how were the tasks evaluated? For example, in the Tesla target, input a comment about Tesla and output a stance, then the comment is a conversation? If it's a conversation, is it about the stance of the conversation as a whole, or about each utterance in the conversation?

(5) According to the statistical data in Table 3, the number of images and the number of text are not equal. Then how does the model handle the absence of image input?

(6) The lack of an ethical statement for the dataset and some copyright issues.

2 For approach, the MLLM-SD framework proposed in this paper is consistent with the existing multimodal large language model structure, and it does not seem to be very novel to impress me.

3 For experiments:

(1) Why not compare with existing multimodal large language models such as LLaVA, MiniGPT4, etc.? I was curious about the performance of existing open source multimodal large language models on MmMtCSD datasets.

(2) How was the result evaluated? I can see that F1 is calculated for both against and favor in Table 6 and 7, but the actual data does not contain three categories: against, favor and none, so none is not taken into account when calculating F1 of the two categories.

(3) GPT4-V is used in the results of multimodal experiments in Table 6, 7and 8. How is GPT4-V evaluated? What is the experimental setup like? Is it a zero-shot?

**Suitability:**

2

---

### Official Review · Reviewer_ey9i · 2024-05-27

**Rating:** 2
**Confidence:** 4

**Summary:**

This paper proposes a new multimodal multi-turn conversational stance detection dataset to compensate for the shortcomings of existing research. And a new multimodal large language model stance detection framework was proposed to learn joint stance representations from text and visual modalities.

**Strengths:**

1. This paper is well written and easy to follow.

2. This paper proposes a dataset for multimodal stance detection, which may promote the development of this field.

**Limitations:**

1. This paper does not provide novel methods for this dataset. For example, the introduction of image captions and the utilization of conversation history information mentioned in this paper have already been extensively studied.

2. The target of this dataset is not innovative, it has only been integrated based on existing work.

3. This dataset consists of visual and textual modalities. So the baseline should compare more pre-trained models of image-text, such as InstructBLIP, MiniGPT-4, LLaVA etc.

4. There are many errors in the citation of references.

**Suitability:**

3

---

### Meta-Review · Area_Chair_1cbA · 2024-07-02

**Recommendation:** Accept (Oral)
**Confidence:** 5

**Metareview:**

This paper introduces a multimodal multi-turn conversation stance detection task, contributing with a new dataset, MmMtCSD, and by a multimodal LLM-based framework to learn joint stance representations from text and images.

Reviewers find the paper to be well-written (ey9i, A95K, Hd8L ) and the dataset contribution to be valuable for the community (ey9i, A95K, QNXi, Hd8L).

Regarding the experimental evaluation, while in general reviewers deemed it to be sufficient (A95K) and solid (Hd8L), several limitations were pointed out, with a common limitation being the lack of comparison with state-of-the-art methods (ey9i, A95K, QNXi, Hd8L).

The paper initially got mixed reviews, with two Weak Accepts, one Borderline Accept, and one Weak Reject. After the rebuttal, the authors managed to address some of the reviewers' concerns, and the paper ended with all reviewers giving a positive score, with two Borderline Accepts, one Weak Accept and one Accept.

Therefore, I agree with the reviewers' final assessment, and suggest this work to be accepted as Poster, provided that authors include the clarifications discussed in their rebuttal in the final manuscript.